# Chinese Herbal Extracts Mitigate Ammonia Generation in the Cecum of Laying Hens: An In Vitro Study

**DOI:** 10.3390/ani13182969

**Published:** 2023-09-20

**Authors:** Miao Li, Kunxian Feng, Jingyi Chen, Tianxu Liu, Yinbao Wu, Jiandui Mi, Yan Wang

**Affiliations:** 1Heyuan Branch, Guangdong Laboratory for Lingnan Modern Agriculture, College of Animal Science, South China Agricultural University, Guangzhou 510642, China; 20213139047@stu.scau.edu.cn (M.L.); 13422174008@163.com (K.F.); buerchen0911@163.com (J.C.); liutianx@cau.edu.cn (T.L.); wuyinbao@scau.edu.cn (Y.W.); mijiandui@163.com (J.M.); 2Guangdong Provincial Key Lab of Agro-Animal Genomics and Molecular Breeding, South China Agricultural University, Guangzhou 510642, China; 3National Engineering Research Center for Breeding Swine Industry, College of Animal Science, South China Agricultural University, Guangzhou 510642, China

**Keywords:** ammonia, laying hen, Chinese herbal extract, astragalus extract, gut microbiota

## Abstract

**Simple Summary:**

This study aimed to investigate the impact of 11 Chinese herbal extracts (cinnamon extract, Osmanthus extract, tangerine peel extract, dandelion extract, Coptis chinensis extract, honeysuckle extract, Pulsatilla root extract, yucca extract, licorice extract, Ginkgo biloba extract, and astragalus extract) on ammonia emissions during in vitro fermentation of the cecum of laying hens. The results showed that the most significant ammonia inhibition was achieved via astragalus extract, resulting in a 26.76% reduction. Astragalus extract inhibited ammonia emission from laying hens by changing the gut microbial community structure, reducing the relative abundance of ammonia-producing bacteria, and reducing microorganisms’ uricase and urease activities.

**Abstract:**

The objectives of the study were to screen one or several Chinese herbal extracts with good ammonia emission reduction effects using an in vitro gas production study. The study consisted of a control (without Chinese herbal extract), and 11 experimental groups with added cinnamon extract (CE), Osmanthus extract (OE), tangerine peel extract (TPE), dandelion extract (DE), Coptis chinensis extract (CCE), honeysuckle extract (HE), Pulsatilla root extract (PRE), yucca extract (YE), licorice extract (LE), Ginkgo biloba extract (GBE), or astragalus extract (AE). The results showed that HE, PRE, YE, LE, GBE, and AE significantly reduced ammonia production (*p* ≤ 0.05). The most significant ammonia inhibition was achieved via AE, resulting in a 26.76% reduction. In all treatments, Chinese herbal extracts had no significant effect on pH, conductivity, or uric acid, urea, and nitrate-nitrogen concentrations (*p >* 0.05). However, AE significantly reduced urease activity and the relative activity of uricase (*p* ≤ 0.05). AE significantly increased the relative abundance of *Bacteroides* and decreased the relative abundance of *Clostridium*, *Desulfovibrio*, and *Prevotell* (*p* ≤ 0.05). Astragalus extract inhibited ammonia emission from laying hens by changing the gut microbial community structure, reducing the relative abundance of ammonia-producing bacteria, and reducing microorganisms’ uricase and urease activities.

## 1. Introduction

Poultry production is quickly expanding, and harmful gases created during rearing, such as ammonia [1], are attracting increasing attention because ammonia emissions from livestock not only contribute to the environment but are also harmful to the health of people and animals. Elevated ammonia concentrations in poultry barns harm animals’ productive performance and induce diseases such as pulmonary edema, tracheitis, dyspnea, anemia, coma, and even suffocation [2,3,4,5]. Moreover, ammonia will react with sulfur dioxide, nitrogen oxides, and other oxidation products in the atmosphere to generate important components of PM_2.5_ [6], such as ammonium nitrate and ammonium sulfate, thus causing environmental pollution [7]. Compared with other livestock and poultry breeds, the highest proportion of feed nitrogen (more than 70%) in laying hens will convert into ammonia [8]. As a result, one of the important concerns relating to the sustainable growth of animal husbandry and raising the standard of living for rural communities is lowering ammonia emissions from laying hens.

There are two main routes of ammonia production in laying hens. The essence of ammonia production in laying hens lies in the decomposition of uric acid and undigested proteins and amino acids in the intestinal tract under the action of microorganisms to produce ammonia [9]. Uric acid accounts for about 80% of the total nitrogen in the excreta of laying hens [10]. After uric acid flows back to the cecum, under the action of uricase and urease secreted by microorganisms, it is decomposed to produce NH_4_^+^ and CO_2_ [11], and NH_4_^+^ is converted into ammonia under alkaline conditions [12]. Among them, about 60–70% of uric acid in the cecum will be degraded, and about 38% will be converted into ammonia [13]. *Escherichia*, *Clostridium*, *Proteus*, and *Klebsiella Trevisan* in the gut have a high urease activity [14]. In addition, ammonia is derived from the fermentation of proteins by microorganisms in laying hens. Due to the short intestine of laying hens, many undigested proteins in the small intestine are hydrolyzed into amino acids and peptides by enzymes secreted by microorganisms [15]. Amino acids and polypeptides are catabolized after entering the cecum; bacteria decompose their metabolites to synthesize bacterial proteins, while ammonia is generated via deamination reactions [16]. However, it is a sluggish and less significant route. Similar to *Clostridium*, *Campylobacter* can cause deamination [17]. Therefore, ammonia emission reduction can be achieved by inhibiting the formation of uric acid, promoting the digestion and absorption of protein and amino acids in laying hens, inhibiting the proliferation of microorganisms related to ammonia production, and reducing the activities of uricase and urease [18].

Chinese herbal extracts possess diverse effective ingredients such as polysaccharides, alkaloids, organic acids, glycosides, and flavonoids [19,20]. At present, there have been many studies on the effects of Chinese herbal medicine extracts on animal performance and immune indexes [21]. Still, the effects on odor emission of poultry production are mainly concentrated on yucca extract and camphor plant extract [22,23]. In addition, there are few studies comparing the effects of different Chinese herbal medicine extracts. The efficiency and mechanism of removing ammonia using Chinese herbal extracts in laying hens, which produce the most ammonia, have yet to be reported. However, the active ingredients in Chinese herbal medicine confer on it the potential to reduce ammonia emissions. The effective components in Chinese herbal extracts can reduce the formation of uric acid by inhibiting the activity of xanthine oxidase [24]. Glycosides, alkaloids, flavonoids, and other active substances in traditional Chinese herbal extracts can also improve the digestibility of feed protein, thereby reducing the concentration of microbial fermentation substrates and reducing ammonia production [25]. A variety of Chinese herbal extracts have broad-spectrum antibacterial effects, which can inhibit the proliferation of ammonia-producing bacteria such as Escherichia coli, Clostridium, and Streptococcus [26,27,28]. Therefore, in order to expand the application of Chinese herbal extracts, this study examined the efficacy and possible mechanisms of ammonia reduction of cinnamon extract (CE), Osmanthus extract (OE), tangerine peel extract (TPE), dandelion extract (DE), Coptis chinensis extract (CCE), honeysuckle extract (HE), Pulsatilla root extract (PRE), yucca extract (YE), licorice extract (LE), Ginkgo biloba extract (GBE), and astragalus extract (AE) using an in vitro study model. In order to further highlight the putative mechanism underlying the impact of Chinese herbal extracts on ammonia emissions, we also examined fermentation parameters and relative microbial abundance. The current study broadens the uses of Chinese herbal extracts in the poultry industry by supplying theoretical support and scientific application guidelines for lowering ammonia emissions in laying hens.

## 2. Materials and Methods

### 2.1. Experimental Design and In Vitro Gas Production

Hyline Grey laying hens at the age of 78-weeks-old were obtained from a local lay hen producer in this investigation. The hens were provided with clean drinking water and a regular corn-and-soybean diet that met their nutritional needs [29]. Throughout the experimental period, in order to regulate the management of the work of the test animals and ensure the quality of the research, the laying hens were managed in accordance with the program approved by the Animal Experimental Committee of South China Agricultural University. Following a period of 28 days of feeding, sixty laying hens with a body weight of 1.78 ± 0.08 kg were selected for slaughter. After the laying hens were killed, the abdominal cavity was opened, and the cecum contents were quickly collected. After being fully mixed, the buffer solution (35 g of NaHCO_3_ plus 4 g of NH_4_HCO_3_ per L) was added at the ratio of 1:3 (*w*/*v*), four layers of gauze were used to filter the discarded filter residue, and the filtrate was collected. The filtrate was placed in a constant temperature water bath at 42 °C, and carbon dioxide (CO_2_) was continuously injected, which constituted the bacteriological liquid.

The inoculant solution was configured as described by Menke and Steingass and modified to be suitable for laying hens. The inoculum solution was then mixed with the above bacterial solution at a volume of 2:1 as an in vitro fermentation solution [30,31]. The piston should evenly apply Vaseline to the surface of the syringe before inserting it, and slowly insert the syringe to prevent the fermentation substrate from blowing out. The experimental group received 0.1% Chinese herbal extract which was mixed evenly, 500 mg of fermentation substrate was accurately weighed for each group, the fermentation substrate was slowly moved into the bottom of the syringe with a long strip of paper, and the prepared in vitro fermentation liquid sucked into the syringe: 30 mL each. The control group did not receive Chinese herbal extract, and the other steps were the same. The syringe was sealed with a water stop clamp, and the fermentation tube was placed in an anaerobic environment and cultured at 60 rpm in a constant temperature oscillator at 42 °C for 12 h.

### 2.2. Sample Collection

Following a 12-h incubation period, the syringes were subjected to a cold bath in order to halt the fermentation process, and the amount of air in the top of each syringe was measured. We immediately transferred the headspace gas to the sulfuric acid absorption solution to collect the ammonia; we immediately took 10 mL of the fermentation solution and placed it in a 15 mL centrifuge tube to determine the pH value. In addition, part of the fermentation liquid was collected and stored in the refrigerator at −80 °C to determine the relative abundance of ammonium-nitrogen, nitrate-nitrogen, uricase, urease, uric acid, urea, and microbial relative abundance.

### 2.3. Sample Analysis

The ammonia collected in the sulfuric acid solution, ammonium-nitrogen, nitrate-nitrogen, uric acid, and urea were measured as described by Wang et al. The determination of urease and uricase activity was conducted by colorimetry, also as described by Wang et al. [31].

In order to assess the impact of eleven distinct Chinese herbal extracts on the diversity of the microbial community, an analysis was conducted on the total bacterial DNA. The DNA was extracted using a bacterial DNA extraction kit (Omega, Norcross, GA, USA). Electrophoresis on a 1% agarose gel was used to check the integrity of the DNA samples. The DNA purity was assessed through the measurement of the absorbance ratio at 260 and 280 nm using an advanced ultraviolet (UV) spectrophotometer. It was necessary for the DNA samples to have an OD260:OD280 ratio within the range of 1.8 to 2.0. The DNA samples were forwarded to Beijing Novogene Co., Ltd. (Beijing, China) for the purpose of conducting DNA analysis.

### 2.4. Statistical Analyses

The data were statistically analyzed and preliminarily sorted by Excel 2010. The data were analyzed by a one-way analysis of variance (ANOVA) using SPSS software (version 22.0, Chicago, IL, USA), and multiple comparisons were made using Duncan’s test. Test data are expressed as the mean ± standard error of the mean. At *p* ≤ 0.05, there was a significant difference. The 16S rRNA data were analyzed with QIIME 1.9.1 [32]. The determination of alpha diversity, PCA plots, the phylum heatmap, and the identification of important genera between treatments was conducted using the Phyloseq software package version 3.8 [33]. The microbiome’s function was estimated through the utilization of PICRUSt, which was afterward followed by the examination of the significance of anticipated functional genes and the generation of plots utilizing DESeq2 version 1.26.0 in the R programming language [34].

## 3. Results

### 3.1. Total Gas and Ammonia Production

Using an in vitro fermentation methodology that has been described as a valid method for imitating gas production from microbial fermentation in the cecum, this study assessed the odorous gas emissions from laying hens. The effects of eleven different types of Chinese herbal extracts on the production of total gas and ammonia in an in vitro fermentation broth are shown in Figure 1. As shown in Figure 1a, compared with the control group, the CE (*p* = 0.002), OE (*p* = 0.000), TPE (*p* = 0.001), DE (*p* = 0.014), CCE (*p* = 0.001), HE (*p* = 0.000), PRE (*p* = 0.000), YE (*p* = 0.018), LE (*p* = 0.004), GBE (*p* = 0.008), and AE (*p* = 0.002) groups had a significantly reduced total gas production (*p* ≤ 0.05). The emission reduction rates were 23.68%, 32.46%, 27.19%, 18.42%, 27.19%, 26.32%, 30.70%, 30.70%, 17.54%, 27.19%, and 20.18%, respectively. Figure 1b shows the effects of eleven different types of Chinese herbal extracts on the concentration of ammonia production during in vitro fermentation. Variance analysis showed no significant difference between the CE (*p* = 0.519), OE (*p* = 0.066), TPE (*p* = 0.152), DE (*p* = 0.152), and CCE (*p* = 0.052) groups and the control group (*p >* 0.05). Compared with the control group, the HE (*p* = 0.026), PRE (*p* = 0.011), YE (*p* = 0.006), LE (*p* = 0.010), GBE (*p* = 0.003), and AE (*p* = 0.000) groups exhibited a significantly reduced ammonia production (*p* ≤ 0.05), and the emission reduction rates were 14.65%, 16.99%, 18.69%, 20.17%, 21.87%, and 26.76%, respectively. The ammonia production in the GBE and AE groups was significantly lower than in the CE, OE, TPE, DE, and CCE groups (*p* ≤ 0.05). The effect of ammonia emission reduction in the AE group was the greatest.

### 3.2. Contents of Ammonium-Nitrogen and Nitrate-Nitrogen

The effects of eleven different types of Chinese herbal extracts on the content of ammonium-nitrogen and nitrate-nitrogen in an in vitro fermentation broth are shown in Figure 2. As shown in Figure 2a, the results of the ANOVA indicated that the concentrations of ammonium-nitrogen in the LE (*p* = 0.018) group and AE (*p* = 0.019) group were significantly lower than that in the OE group (*p* ≤ 0.05), and there was no significant difference in the concentration of ammonium-nitrogen between the other groups (*p >* 0.05). The addition of Chinese herbal extracts had no significant effect on the change in nitrate-nitrogen concentration in the fermentation broth (*p >* 0.05).

### 3.3. Contents of Uric Acid and Urea

Figure 3 illustrates the impact of eleven distinct Chinese herbal extracts on the levels of uric acid and urea in an in vitro fermentation broth. The analysis of ANOVA results indicated that there were no statistically significant differences in the levels of uric acid and urea in the fermentation broth between each treatment group and the control group (*p >* 0.05).

### 3.4. Relative Activity of Uricase and Urease

The effects of eleven different types of Chinese herbal extracts on the relative activity of uricase and urease activity in an in vitro fermentation broth are shown in Figure 4. As shown in Figure 4a, compared with the control group, the HE (*p* = 0.018), PRE (*p* = 0.006), YE (*p* = 0.005), LE (*p* = 0.049), GBE (*p* = 0.019), and AE (*p* = 0.001) groups exhibited a significantly reduced uricase activity (*p* ≤ 0.05), and the RPE, YE, and AE groups exhibited a significantly lower uricase activity than the CE, OE, TPE, DE, and CCE groups (*p* ≤ 0.05). As shown in Figure 4b, compared with the control group, the HE (*p* = 0.000), PRE (*p* = 0.002), YE (*p* = 0.004), GBE (*p* = 0.000), and AE (*p* = 0.000) groups exhibited a significantly reduced urease activity in the fermentation broth (*p* ≤ 0.05). The GBE and AE groups showed significantly lower activities than the HE, PRE, and YE groups (*p* ≤ 0.05).

### 3.5. The pH and Electrical Conductivity

The effects of eleven different types of Chinese herbal extracts on the pH and electrical conductivity in an in vitro fermentation broth are shown in Figure 5. The results of the ANOVA showed that the pH and electrical conductivity in the fermentation broth of each treatment group were not significantly different from those of the control group (*p >* 0.05).

### 3.6. Microbial Diversity and Relative Abundance

The methodology employed in this study involved the utilization of 16S rRNA sequencing to assess both the abundance and structure of the microbiome in response to the various therapies. The effects of eleven different types of Chinese herbal extracts on the microbial diversity and relative abundance in the fermentation broth in vitro are shown in Figure 6. The alpha diversity of the microbial community in the cecum of laying hens was represented by the chao1 index. The chao1 index reflects the abundance of species in a sample, that is, taking into account only the number of species in the sample rather than the abundance of each species in the sample. Compared with the control group, TPE, YE, and LE significantly increased the number of the chao1 index (*p* ≤ 0.05). The bacterial microbiota composition differed between different processing groups, as shown by the principal coordinates analysis (PCoA) plot (Figure 6b).

The variation in the relative abundance of microorganisms in the fermentation broth at the phyla level is shown in Figure 6c. Compared with the control group, GBE significantly increased the relative abundance of Firmicutes (*p* ≤ 0.05). YE and GBE significantly decreased the relative abundance of Bacteroidetes (*p* ≤ 0.05). AE significantly decreased the relative abundance of Proteobacteria (*p* ≤ 0.05).

The variation in the relative abundance of microorganisms in the fermentation broth at the genus level is shown in Figure 6d. At the genus level, 78, 83, 80, 82, 81, 87, 83, 85, 79, 77, 79, and 82 bacteria were identified in the control group, CE, OE, TPE, DE, CCE, HE, PRE, YE, LE, GBE, and AE groups, respectively. The top 15 genera with relative abundance were selected for analysis. Compared with the control group, HE, PRE, and AE significantly increased the relative abundance of *Bacteroides* (*p* ≤ 0.05). The results showed that HE and GBE could dramatically increase the relative abundance of *Flavonifractor* (*p* ≤ 0.05). PRE and AE significantly decreased the relative abundance of *Desulfovibrio* (*p* ≤ 0.05). YE, GBE, and AE decreased the relative abundance of *Prevotella* in the fermentation broth (*p* ≤ 0.05). HE, PRE, LE, and AE reduced the relative abundance of *Clostridium* in the fermentation broth (*p* ≤ 0.05).

### 3.7. Microbial Function Prediction Analysis

PICRUSt was employed to ascertain the functional attributes of the microbiome, relying on the analysis of 16S rRNA sequencing data. The KEGG database serves as an integrated platform for the integration of genomic, chemical, and system function information [35]. There were 6909 KEGG orthology (KO) pathways noted in the results of this study. Then, 59 KO IDs associated with ammonia production were filtered from the prediction results. A total of 20 KO IDs showed a greater abundance in the GBE treatment, with examples including hydroxylamine reductase, carbamate kinase, ferredoxin-nitrite reductase, nitrogenase molybdenum-iron protein beta chain, and the nitrogenase molybdenum-cofactor synthesis protein NifE (Figure 7a). Further, glutamate dehydrogenase (NADP+), hydroxylamine reductase, nitrogenase iron protein NifH, dihydrolipoamide dehydrogenase, glutaminase, nitrogenase molybdenum-iron protein beta chain, the nitrogenase molybdenum-cofactor synthesis protein NifE, nitrogenase molybdenum-iron protein alpha chain, urease alpha subunit, and urease accessory protein were elevated in the AE treatment (Figure 7b).

## 4. Discussion

The livestock industry has a heavy focus on ammonia emissions as an environmental problem. Chinese herbal medicine extracts have the advantages of natural sources, no pollution, low cost, etc., and are rich in a variety of active substances and have a wide range of effects, which may have the potential to reduce ammonia gas in laying hens. Yucca extract has been proven to lower the levels of ammonia [22,36,37,38]. Thus, 11 Chinese herbal extracts were selected to validate their potential to minimize ammonia emissions from laying hens during in vitro fermentation. The results of this study demonstrated that HE, PE, YE, LE, GBE, and AE could significantly reduce ammonia production in the cecum layer, among which GBE and AE were superior. Adding licorice extract could decrease ammonia production without affecting the overall rumen fermentation process [39]. Studies have shown that the addition of licorice extract reduces the ammonia concentration at 12 h of fermentation [40]. The results showed that adding saponins could reduce the concentration of ammonia-N during in vitro fermentation, and the concentration of ammonia-N decreased in a dose-dependent manner [41]. Another study showed that capsicum, coriander, and thyme decreased ammonia-N concentration during in vitro batch microbial fermentation [42].

Due to the different types and contents of active components in Chinese herbal extracts and the complex mechanism of action, the effects of different types of Chinese herbal extracts are inconsistent. Generally speaking, the reason why Chinese herbal extracts affect the production of ammonia in animals may be related to their effects on intestinal flora, uricase, and urease activities [43]. Firstly, the activities of uricase and urease may be related to the production of ammonia in animals. The higher the uricase and urease activities, the more ammonia gas is produced by gut microbes [44]. The differences in the concentrations of uric acid and urea in the fermentation broth were insignificant, possibly because urea was in a dynamic equilibrium of urea production from uric acid degradation and urea decomposition. Shi et al. claimed that Magnolia officinalis and Cassia obtusifolia exhibited significant urease-inhibition actions [45]; Fan et al. showed that yucca extract reduces uricase and urease activity in intestinal contents and reduces ammonia emissions [46]. Our findings are similar to those of these studies. Our research revealed that HE, PRE, YE, LE, GBE, and AE, which may reduce the emission of ammonia gas in laying hens, significantly reduced uricase activity; among them, HE, PRE, YE, GBE, and AE also significantly reduced urease activity. In traditional Chinese medicine extracts, the substances that inhibit uricase and urease are mainly alkaloids, polysaccharides, flavonoids, saponins, terpenes, and phenols. At the same time, GBE is rich in flavonoids and phenols, terpenes, and other active substances, and AE is rich in alkaloids, polysaccharides, flavonoids, and saponins. These results were consistent with the results that GBE and AE exhibited a significantly better inhibition of urease activity than other Chinese herbal extracts. This may be one of the reasons why the ammonia emission reduction effect of these two Chinese herbal extracts was greater than that of other Chinese herbal extracts.

The structure of the animal intestinal flora is closely related to the emission of ammonia [16]. At present, there are many research works on the bacteria related to ammonia production. Ammonia-producing bacteria mainly use amino acids as raw materials for protein synthesis, and ammonia production is mainly produced by deammonification or transammonification. Ammonia-producing bacteria can affect the production of ammonia gas by affecting the action of microbial enzymes and the production of substrates. Scholars have reported mixed results when determining which microbes are the primary producers of ammonia. Mafra et al. found that *Propionibacterium, Clostridium, Streptococcus, Staphylococcus*, and *Bacillus* were common protein- and amino-acid-decomposing bacteria that can produce ammonia by breaking down proteins and amino acids [47]. Macfarlane et al. believed that the microorganisms with a high proteolytic activity were mainly *Bacteroides*, *Clostridium*, *propionibacterium*, *Clostridium*, and *Streptococcus* [17]. The results of the principal component analysis of the samples in this study (Figure 6b) all showed that the additional treatment of different Chinese herbal extracts could affect the microbial community structure of the in vitro fermentation system. Under the conditions of this experiment, HE, PRE, LE, and AE reduced the relative abundance of *Clostridium* in the fermentation broth (*p* ≤0.05). *Clostridium* is considered to have strong ammonia-producing properties. YE, GBE, and AE decreased the relative abundance of *Prevotella* in the fermentation broth (*p* ≤ 0.05). *Prevotella* are highly ammonia-producing bacteria that produce ammonia gas by fermenting proteins and amino acids [48]. HE, PRE, and AE significantly increased the relative abundance of *Bacteroides* (*p* ≤ 0.05). The ammonia production capacity of *Bacteroides* is weak, and it mainly ferments carbohydrates for energy and rarely uses nitrogen compounds such as amino acids [49]. HE, PRE, and AE reduced the relative abundance of *Clostridium* and urease activity in the fermentation broth (*p* ≤ 0.05). Studies have also shown that *Clostridium*, *Proteus*, and *Klebsiella* have a high urease activity and can decompose urea to produce ammonia [50]. The decrease in the relative abundance of highly ammonia-producing bacteria may be caused by the inhibition of urease activity synthesized by itself and the inability to decompose and produce ammonia to maintain a stable living environment. Our study showed that supplementation of Chinese herbal extract increased the carbohydrate-fermenting bacteria, such as *Bacteroides*, decreased the bacteria with a higher urease activity, such as *Clostridium*, and increased the bacteria that are less capable of producing ammonia, such as *Bacteroides*. The aforementioned information leads to the conclusion that one route by which a Chinese herbal extract reduces ammonia generation is by changing the microbial populations to favor carbohydrates and reducing nitrogenous fermentation. The second mechanism by which Chinese herbal extracts reduce ammonia production is to reduce urease activity by reducing the abundance of bacteria with a higher urease activity, thereby reducing ammonia production and emissions. The third mechanism by which Chinese herbal extracts reduce ammonia production is to reduce the production of uric acid by inhibiting the activity of uricase, thereby reducing the production and emission of ammonia.

In this study, the Chinese herbal extracts with the best ammonia emission reduction effect in laying hens were selected from 11 kinds of Chinese herbal extracts through an in vitro fermentation model, which provided a theoretical basis for the future application of Chinese herbal extracts in laying hens. This offered a theoretical framework for applying Chinese herbal extracts in laying hens in the future. The in vitro intestinal fermentation model is typically made up of a microbial medium, a gas volume or pressure measuring device, a constant temperature oscillation device, and a fermentation device. It is the perfect tool for the investigation of intestinal post-fermentation in animals with a single stomach because of its benefits of ease of use, high reproducibility, mass manufacturing, and regulation. This test method has been used to measure gas emissions from livestock [31,44]. In vitro intestinal fermentation models for inoculation and colonization currently come in various designs and configurations. However, every in vitro culture method has limitations; thus, it is impossible to completely simulate the microecological environment in vivo. Additionally, commonly questioned and criticized are the repeatability and functional stability of the gut microbiota during in vitro intestinal fermentation models. The technical details, fermentation substrates, and gas production equipment employed in various laboratories vary, so the variability of data between different laboratories also needs to be further explored. To obtain more accurate in vivo results, the results of in vitro models need to be supplemented with in vivo models, which can not only enhance the overall validity of the in vitro model but also distinguish the functional relationship between the gut microbiome and the animal body. Considering the importance of in vivo models, it is suggested to feed the Chinese herbal extract as a feed additive to further clarify the emission reduction effect and influence mechanism of Chinese herbal extracts on ammonia in laying hens.

## 5. Conclusions

In conclusion, the results of our investigation demonstrated that HE, PRE, YE, LE, GBE, and AE considerably reduced ammonia production via the in vitro fermentation of cecal contents in laying hens. Of all the treatment groups, GBE and AE exhibited the best ammonia emission reduction effect, and the emission reduction ratios were 21.87% and 26.76%, respectively. Due to the different types and contents of active components in Chinese herbal extracts and the complex mechanisms of action, the effects of different Chinese herbal extracts are inconsistent. This study further suggested that the reason why AE and GBE affect the production of ammonia in animals may be related to their effects on intestinal flora, uricase, and urease activities. These findings suggest that AE and GBE have a significant impact on lowering ammonia emissions and odor pollution brought on by the poultry industry.

## Figures and Tables

**Figure 1 animals-13-02969-f001:**
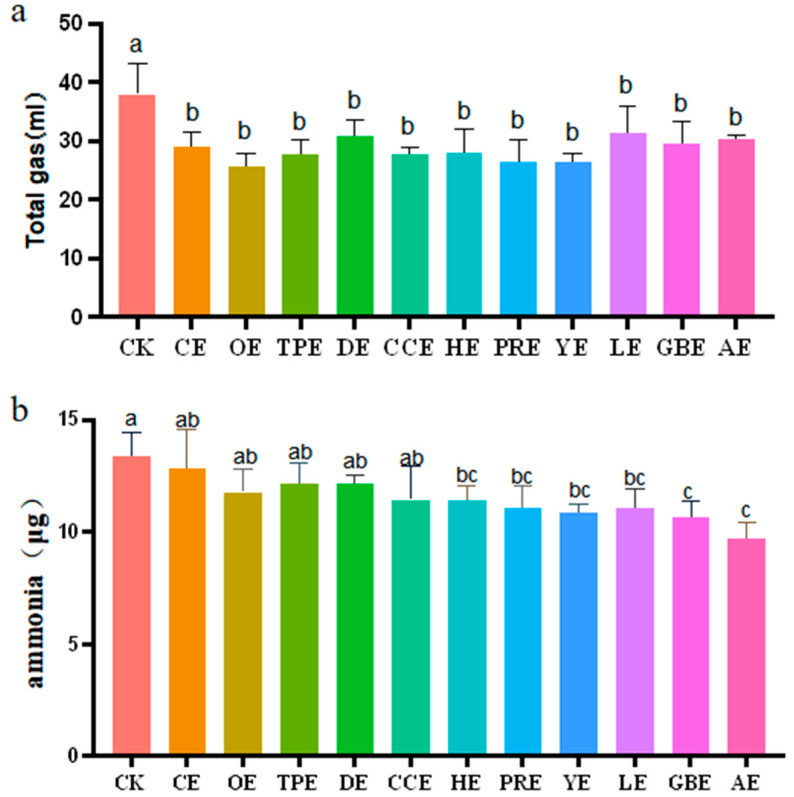
(**a**) Effect of different types of Chinese herbal extracts on the in vitro production of total gas. (**b**) Effect of different types of Chinese herbal extracts on the in vitro production of ammonia. CK—control group, CE—cinnamon extract, OE—Osmanthus extract, TPE—tangerine peel extract, DE—dandelion extract, CCE—Coptis chinensis extract, HE—honeysuckle extract, PRE—Pulsatilla root extract, YE—yucca extract, LE—licorice extract, GBE—Ginkgo biloba extract, AE—astragalus extract. Error bars indicate standard errors (*n* = 6). Different letters above the bars indicate statistically significant differences between the samples (ANOVA followed by Duncan’s test, *p* ≤  0.05).

**Figure 2 animals-13-02969-f002:**
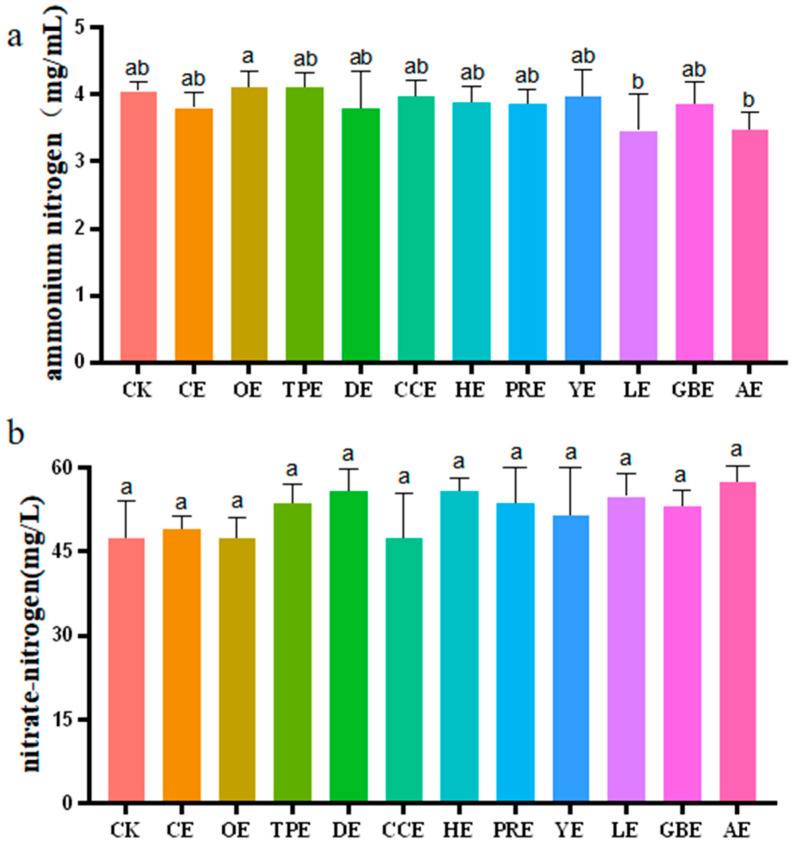
(**a**) Effect of different types of Chinese herbal extracts on the in vitro ammonium-nitrogen content. (**b**) Effect of different types of Chinese herbal extracts on the in vitro nitrate-nitrogen content. CK—control group, CE—cinnamon extract, OE—Osmanthus extract, TPE—tangerine peel extract, DE—dandelion extract, CCE—Coptis chinensis extract, HE—honeysuckle extract, PRE—Pulsatilla root extract, YE—yucca extract, LE—licorice extract, GBE—Ginkgo biloba extract, AE—astragalus extract. Error bars indicate standard errors (*n* = 6). Different letters above the bars indicate statistically significant differences between the samples (ANOVA followed by Duncan’s test, *p* ≤  0.05).

**Figure 3 animals-13-02969-f003:**
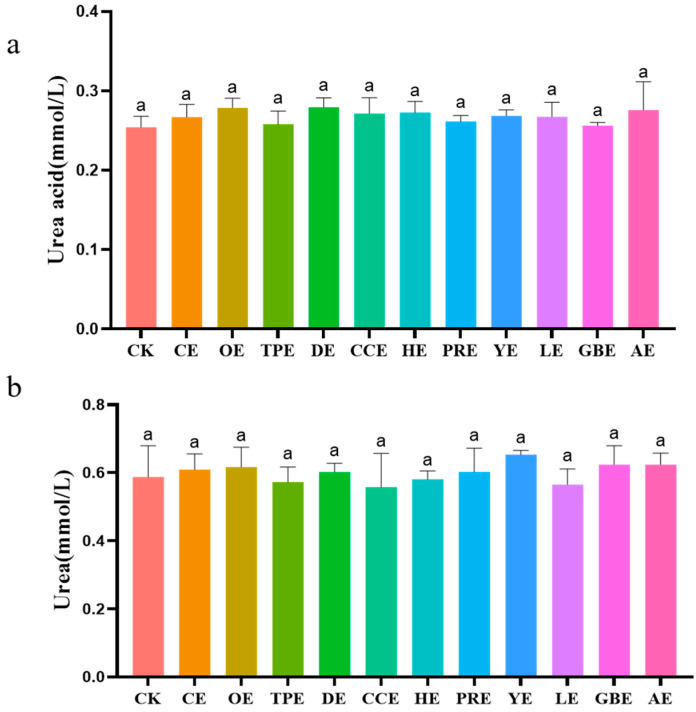
(**a**) Effect of different types of Chinese herbal extracts on the in vitro urea acid content. (**b**) Effect of different types of Chinese herbal extracts on the in vitro urea content. CK—control group, CE—cinnamon extract, OE—Osmanthus extract, TPE—tangerine peel extract, DE—dandelion extract, CCE—Coptis chinensis extract, HE—honeysuckle extract, PRE—Pulsatilla root extract, YE—yucca extract, LE—licorice extract, GBE—Ginkgo biloba extract, AE—astragalus extract. Error bars indicate standard errors (*n* = 6). Different letters above the bars indicate statistically significant differences between the samples (ANOVA followed by Duncan’s test, *p* ≤  0.05).

**Figure 4 animals-13-02969-f004:**
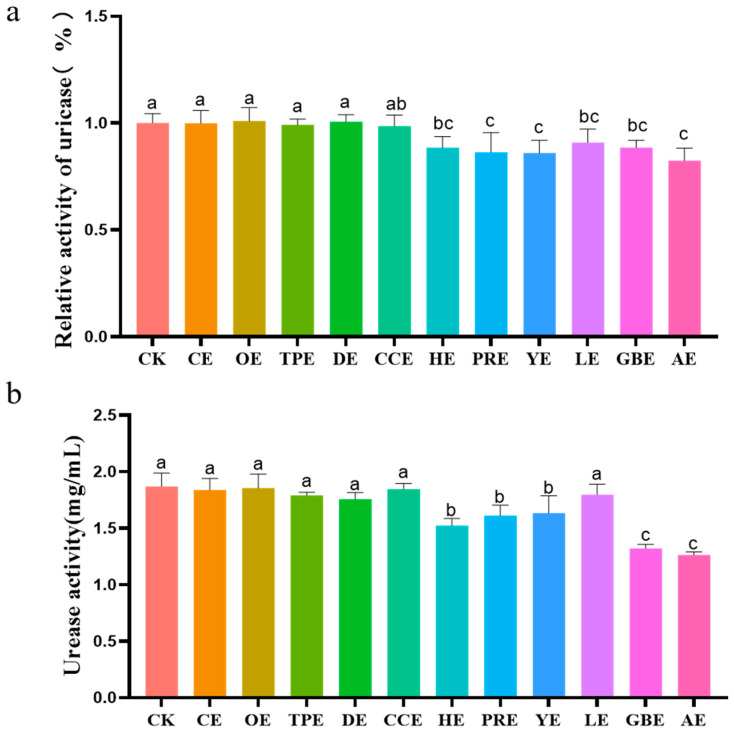
(**a**) Effect of different types of Chinese herbal extracts on the in vitro relative activity of uricase. (**b**) Effect of different types of Chinese herbal extracts on in vitro urease activity. CK—control group, CE—cinnamon extract, OE—Osmanthus extract, TPE—tangerine peel extract, DE—dandelion extract, CCE—Coptis chinensis extract, HE—honeysuckle extract, PRE—Pulsatilla root extract, YE—yucca extract, LE—licorice extract, GBE—Ginkgo biloba extract, AE—astragalus extract. Error bars indicate standard errors (*n* = 6). Different letters above the bars indicate statistically significant differences between the samples (ANOVA followed by Duncan’s test, *p* ≤  0.05).

**Figure 5 animals-13-02969-f005:**
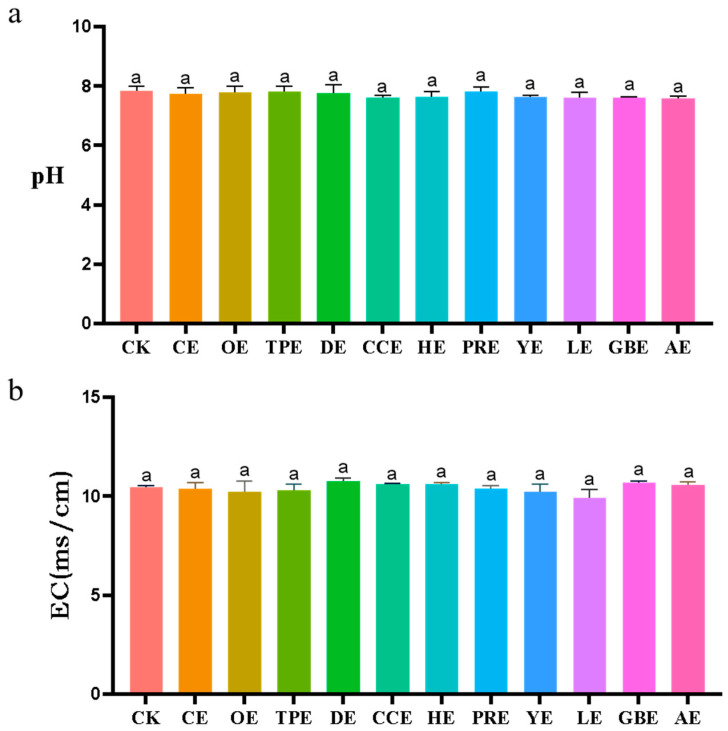
(**a**) Effect of different types of Chinese herbal extracts on the in vitro pH. (**b**) Effect of different types of Chinese herbal extracts on in vitro electrical conductivity. CK—control group, CE—cinnamon extract, OE—Osmanthus extract, TPE—tangerine peel extract, DE—dandelion extract, CCE—Coptis chinensis extract, HE—honeysuckle extract, PRE—Pulsatilla root extract, YE—yucca extract, LE—licorice extract, GBE—Ginkgo biloba extract, AE—astragalus extract. Error bars indicate standard errors (*n* = 6). Different letters above the bars indicate statistically significant differences between the samples (ANOVA followed by Duncan’s test, *p* ≤  0.05).

**Figure 6 animals-13-02969-f006:**
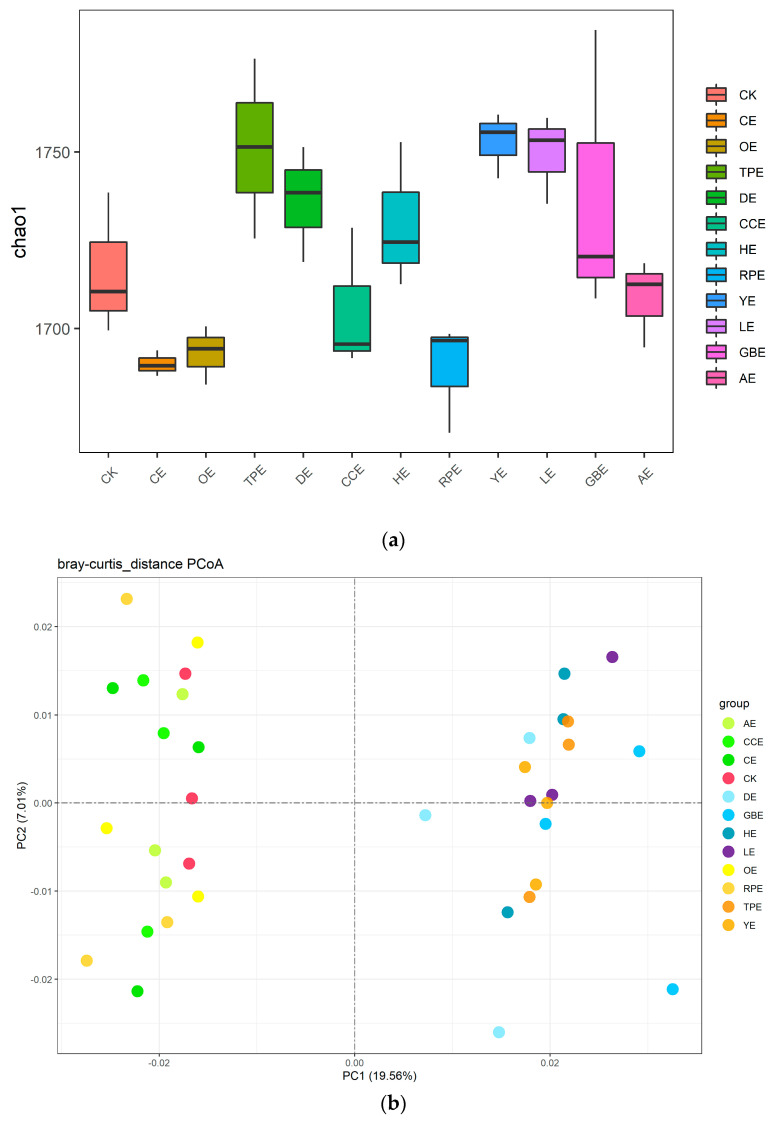
(**a**) Effect of different types of Chinese herbal extracts on Chao 1 index of fermentation bacteria. (**b**) Principal coordinate analyses based on unweighted Unifrac distances. (**c**) Effect of different types of Chinese herbal extracts on the phyla of the microbial community. (**d**) Effect of different types of Chinese herbal extracts on the genera of the microbial community. CK—control group, CE—cinnamon extract, OE—Osmanthus extract, TPE—tangerine peel extract, DE—dandelion extract, CCE—Coptis chinensis extract, HE—honeysuckle extract, PRE—Pulsatilla root extract, YE—yucca extract, LE—licorice extract, GBE—Ginkgo biloba extract, AE—astragalus extract.

**Figure 7 animals-13-02969-f007:**
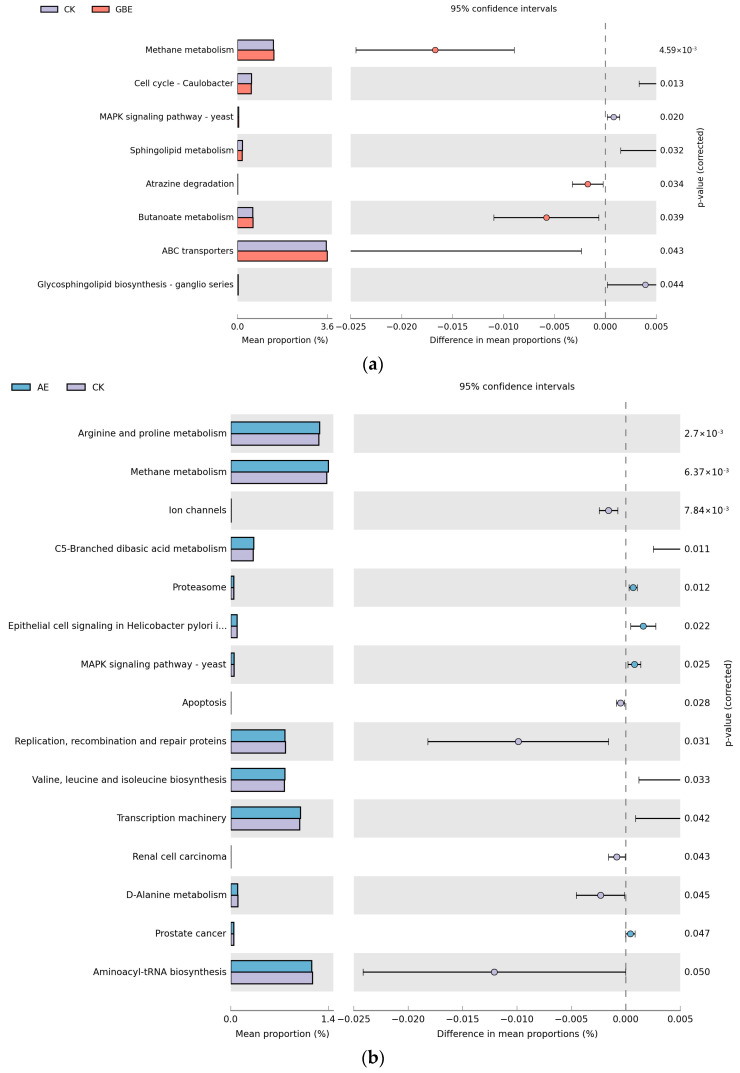
(**a**) Microbial function prediction analysis between CK and GEB. (**b**) Microbial function prediction analysis between CK and AE. CK—control group, AE—astragalus extract, GBE–Ginkgo biloba extract.

## Data Availability

The data presented in this study are available in the Appendix A and NCBI Sequence Read Archive (SRA) repository (https://www.ncbi.nlm.nih.gov/bioproject/PRJNA987521, accessed on 26 June 2023), accession number PRJNA987521.

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
