# Peer review of "Chinese Herbal Extracts Mitigate Ammonia Generation in the Cecum of Laying Hens: An In Vitro Study"

_animals, 2023, doi:10.3390/ani13182969_

Round 1
Reviewer 1 Report
Need to improve materials and methods. Please check the file for comments.

Author Response
请看附件。

Reviewer 2 Report
I recommend major revision because the bibliography is missing which is a key element to study and review an article.
The introduction is not dressed well and needs improvement in language editing. For example, we never start a sentence with “and” L 79.
L 42-65: The text is quite difficult; I think it would be better to create a figure that explains the two mentioned main ways.
L 96: Please clarify what kind of production line is the specific one, that keeps animals 78 weeks old.
Improve the resolution of figures No 6 and 7, and increase the size of the letters in the figures.
Figures 6 & 7. Please use italics.
Lines 319-372: Most of the discussion is a general text that could well be included in the introduction and essentially the management of the results of the present study starts from line 361. I consider it unacceptable and there should be a complete reformation of the text and the discussions, that negotiate the results of the present study.
Reviewer 3 Report
This is a preliminary study te explore the effects of essential oils to modifiy in vitro fermentation with a single caecum digesta content from hens. However, results are far of being representative of their response in live animals. The study should be complemented with in vivo trials to confirm if a compound like this arrive to the hindgut and influence microbiota and fermentation. Moreover, responses in vitro were not significantly different of control diets. Then, additional efforts are requiered in order to reinforce the value of these results.
Round 2
Reviewer 1 Report
All my concerns are satisfactory addressed; except two concerns.
1. Author did not mention the total number of hens.
2. Write P≤0.05 instead of P<0.05

Reviewer 2 Report
The references are quite old in most parts of the article, so please make an effort to put references of the last five years
Reviewer 3 Report
X
